# Secretory Immunoglobulin A Immunity in Chronic Obstructive Respiratory Diseases

**DOI:** 10.3390/cells11081324

**Published:** 2022-04-13

**Authors:** Charlotte de Fays, François M. Carlier, Sophie Gohy, Charles Pilette

**Affiliations:** 1Pole of Pneumology, ENT, and Dermatology, Institute of Experimental and Clinical Research, Université Catholique de Louvain, 1200 Brussels, Belgium; charlotte.defays@saintluc.uclouvain.be (C.d.F.); francois.carlier@chuuclnamur.uclouvain.be (F.M.C.); sophie.gohy@saintluc.uclouvain.be (S.G.); 2Department of Pneumology, CHU UCL Namur, Site Mont-Godinne, 5530 Yvoir, Belgium; 3Lung Transplant Centre, CHU UCL Namur, Site Mont-Godinne, 5530 Yvoir, Belgium; 4Department of Pneumology, Cliniques Universitaires Saint-Luc, 1200 Brussels, Belgium; 5Cystic Fibrosis Reference Centre, Cliniques Universitaires Saint-Luc, 1200 Brussels, Belgium

**Keywords:** immunoglobulin A, mucosal immunity, pIgR, COPD, asthma, cystic fibrosis

## Abstract

Chronic obstructive pulmonary disease (COPD), asthma and cystic fibrosis (CF) are distinct respiratory diseases that share features such as the obstruction of small airways and disease flare-ups that are called exacerbations and are often caused by infections. Along the airway epithelium, immunoglobulin (Ig) A contributes to first line mucosal protection against inhaled particles and pathogens. Dimeric IgA produced by mucosal plasma cells is transported towards the apical pole of airway epithelial cells by the polymeric Ig receptor (pIgR), where it is released as secretory IgA. Secretory IgA mediates immune exclusion and promotes the clearance of pathogens from the airway surface by inhibiting their adherence to the epithelium. In this review, we summarize the current knowledge regarding alterations of the IgA/pIgR system observed in those major obstructive airway diseases and discuss their implication for disease pathogenesis.

## 1. Introduction

Each breath carries thousands of particles towards the airways, constituting many potential threats to lung integrity. Airway mucosal immunity provides both innate and adaptive responses against these inhaled agents, providing inflammatory responses against harmful antigens and tolerogenic mechanisms towards innocuous ones. Failure to these duties may lead to increased antigen penetration and recurrent infections or exaggerated immune responses towards harmless antigens, both situations potentially resulting in chronic airway inflammation. Immunoglobulin (Ig) A represents the predominant Ig at mucosal surfaces in humans [1], where it predominantly lays in its secretory form (S-IgA). Carried towards the airway lumen by its epithelial transporter, the polymeric Ig receptor (pIgR), S-IgA plays a crucial role in the immune exclusion of inhaled pathogens while IgA also regulates immune cells residing in or attracted to mucosal tissues [2].

The pIgR/IgA system is altered in several chronic respiratory diseases, such as chronic obstructive pulmonary disease (COPD), asthma, and cystic fibrosis (CF). Asthma and COPD are frequent diseases, with a worldwide prevalence of 262 and 212 million people, respectively [3]. While COPD mainly affects adults over 65 years old, asthma also affects children and young adults [3]. CF affects at least 100, 000 people worldwide and is the most common lethal autosomal recessive Caucasian disease, affecting up to 1:1353 new-born in Ireland, the highest prevalence in Europe [4]. Although driven by distinct pathogenic mechanisms, that are further described, these respiratory diseases share common features characterized by chronic airway obstruction and the occurrence of acute episodes of flare-ups called exacerbations [5]. These episodes of acute increased respiratory symptoms are associated with increased medication use, emergency department visits, hospitalization, and death [6,7,8,9]. Viral and bacterial triggers account for a substantial number of these episodes [10]. Numerous factors are involved in mucosal defence, such as epithelial barrier integrity, muco-ciliary clearance, or antimicrobial molecules, along with the pIgR/IgA system, which is altered in these chronic respiratory diseases. Importantly, serum IgA deficiency has been associated with higher exacerbation risk in COPD [11] as well as with increased incidence of allergic diseases [12].

In this review, we summarize the current knowledge regarding the dysregulation of pIgR/IgA mucosal immunity, focusing on the respiratory system. We will then develop how the dysregulation of pIgR/IgA mucosal immunity could impact the pathogenesis and clinical course of chronic airway diseases such as COPD, asthma, and CF.

## 2. The Mucosal S-IgA System in Homeostasis

The mucosal barrier system is constitutively made of several components. The epithelial layer and the intercellular junctions, with the mucus layer on top, and the lamina propria underneath [13,14], form a physical barrier against exogenous antigens and pathogens. Mucosal antimicrobial molecules, cellular innate immunity and mucosal adaptative immunity are complementary lines of defence against aggressions [15]. In the bronchi, the epithelium is pseudostratified, and composed by numerous cell types whose proportions are tightly controlled [16,17]. Most of the bronchial airway epithelium consists of ciliated and goblet cells that together constitute the mucociliary elevator clearing particulates and other irritants out of the airspaces. As well as these cell types, the airway epithelium also comprises basal, club and neuroendocrine cells, along with rare ionocytes [18]. Ciliated cells are prominent and represent more than 50% of all airway epithelial cells. They possess around 300 cilia whose synchronized beating pushes the mucus layer towards the trachea and the larynx [19]. Goblet cells, accounting for 5 to 15% of airway epithelial cells in health, produce mucus and are virtually not present in small airways [19]. Basal cells are multipotent stem cells that both anchor the epithelium to the underlying lamina propria and drive epithelial homeostasis and orderly regeneration after injury [20,21]. They represent 5 to 30% of epithelial cells, their proportion decreasing from the trachea down to the respiratory bronchioles [22]. Club cells are dome-shaped cells involved in host defence, and represent 20% of epithelial cells in small airways, where they also behave as progenitor cells [23,24]. Neuroendocrine cells are rare, innervated cells (< 1% of airway epithelial cells) [25] that are thought to be involved in oxygen sensing, smooth muscle tonus and immune responses [26,27]. Finally, recently discovered ionocytes seem to control the airway surface liquid and mucus viscosity [28]. Among these cell types, ciliated, goblet and club cells have been robustly shown to express the polymeric immunoglobulin receptor (pIgR) and therefore participate in the epithelial transcytosis of dimeric IgA (d-IgA) towards the apical mucus layer (see below) [29,30], while recent single cell transcriptomic data and the emergence of the Human Lung Atlas suggest that ionocytes (but not basal and neuroendocrine cells) also express the pIgR [31].

### 2.1. Production and Structure of S-IgA and pIgR

IgA constitutes the most prevalent Ig isotype at mucosal sites and the second most prevalent in serum after IgG. Therefore, IgA is the most abundantly produced Ig in the human body, with an average production rate of 66 mg/kg/day [32,33]. IgA largely mediates the adaptive humoral immune defence at mucosal surfaces [34,35], while its role in serum remains relatively unexplored. IgA is produced by B cells both in the systemic and mucosal immune systems, the latter being referred to as mucosa-associated lymphoid tissue (MALT) which includes mucus layers, epithelial cells, lymphoid tissues and immune molecules of the mucosal lamina propria [36]. Structurally, IgA is found as monomers (m-IgA) or polymers, mainly consisting of dimers (d-IgA), although some larger forms also exist. M-IgA is composed of two light chains κ and λ, common to all types of Ig’s, covalently linked to two specific heavy chains α [37].

In serum, IgA represents 6 to 15% of total immunoglobulins [38], and predominates as monomers (85% to 90%). Serum IgA mainly originates from the bone marrow, although the spleen and peripheral lymph nodes could contribute to a lower extent [38]. In the mucosal lamina propria, plasma cells mainly produce IgA as dimers (d-IgA) consisting of two IgA monomers joined to the joining (J) chain at the Fc region by a disulfuric bridge [33,39]. Subepithelial d-IgA molecules can bind to the polymeric immunoglobulin receptor (pIgR) at the basolateral pole of the epithelium [34], ensuring the transcytosis of pIgR/d-IgA complexes across the epithelium. Following transcytosis, the ectodomain of pIgR (called secretory component, SC) is cleaved, allowing the apical release of d-IgA/SC complexes that constitute secretory (S-)IgA [33,35]. S-IgA is the predominant Ig in mucosal secretions such as breast milk, colostrum, tears, saliva, digestive as well as respiratory and genitourinary tract secretions [32]. S-IgA is predominantly found in a dimeric form, although some larger polymers (mainly tetramers) have also been identified [32].

In humans, two IgA subclasses exist, namely IgA_1_ and IgA_2_. Each of these IgA subtypes can be found in a monomeric, dimeric, or secretory form [40]. In bronchial secretions, IgA_2_ accounts for 30% of total IgA, compared to 10% in plasma [40]. This strongly suggests that there is a local production of IgA in the lung, not only spilling from the blood vessels, but also there is a substantial level of IgA2 secreting B cells within the bronchial mucosae [41].

B cells that secrete IgA in the airways are antibody-secreting plasma cells that are generated following cognate interactions between T cells and dendritic cells that have taken up and processed a specific antigen [42,43]. These conventional B(2) cells undergo class switch recombination towards IgA upon a TGF-β-dependent mechanism [44] and produce high affinity IgA antibodies. Alternatively, B1 cells are B cells that produce so-called “natural” IgM and IgA antibodies although those B cells have mainly been described in the serous cavities and in the gut [45]. Thus, the nature and origin of IgA^+^ B cells in the lung, as well as their regulation, are poorly characterized. Blood B cells are defined into five subsets, namely transitional, naïve, switched memory, unswitched memory and plasma cells, but similar maturation subsets have yet to be defined in the lung [46]. Nevertheless, it is known from studies in bronchial biopsies of patients with asthma that there is a local IgE production with somatic hypermutation within the airway mucosa [47]. Since IgA_2_ is the only possible switch for IgE^+^ B cells, as IgA2 heavy chain constant region gene is the only gene located downstream of the IgE gene [48], the hypothesis that IgE^+^ B cells could switch to IgA_2_ in the lung has been evoked, and supported by the observation that allergen immunotherapy may induce the induction of antigen-specific IgA_2_ antibodies in allergic patients [49].

pIgR is an 81 kDa transmembrane protein [50,51] that fulfils two main functions. First, it transports polymeric Ig’s (mainly d-IgA and pentameric IgM) across the mucosal epithelium [51]. Second, it serves as a precursor for SC, that is generated after endoproteolytic cleavage of its extracellular part at the apical pole of epithelial cells [52]. The extracellular, transmembrane and cytoplasmic portions of pIgR are respectively constituted by 620, 23, and 103 amino-acids [50]. The extracellular part is divided into six domains, with domains one to five being Ig-homologous domains, and domain six being a non-homologous domain and possibly representing the site of the proteolytic cleavage [50]. Therefore, SC is composed of the five Ig-like domains of pIgR [50,52].

### 2.2. Transcytosis of d-IgA and Functions of S-IgA

pIgR-mediated polymeric Ig transcytosis requires four well-orchestrated steps [51]. First, d-IgA (or pentameric IgM) binds to pIgR through a non-covalent interaction between Ig’s J chain and the extracellular part of pIgR. Second, pIgR is internalized in the clathrin-mediated endocytosis process and delivered in basolateral early endosomes, then in common endosomes. Endocytosed pIgR is then either recycled back to the basolateral membrane, or delivered to the apical surface [50]. Third, the extracellular part of the receptor (SC, bound to the transported Ig), undergoes endoproteolytic cleavage by a host serine proteinase. Lastly, S-IgA consisting of d-IgA and SC, crosses the mucus layer by diffusion. Of note is that, unbound pIgR may also undergo epithelial transcytosis and proteolytic cleavage, thereby releasing free SC that may be found in mucosal secretions [50,51].

At mucosal surfaces, S-IgA functions include neutralization, as well as other biological effects through interactions with, and regulation of, immune cells (via specific Fcα receptors) and microbiota components (Figure 1). The first-line mucosal defence is primarily exerted by S-IgA through its binding of soluble or particulate antigens, preventing their adherence to the epithelium. This non-specific immunity process called ‘immune exclusion’ consists of several sub-steps, including agglutination and entrapment as well as clearance at mucosal surfaces [2,53]. First, agglutination occurs when S-IgA non-covalent binding to microorganisms promotes the formation of macroscopic clumps of pathogens [2,36], altering pathogen physiology and gene expression, depending on the epitope recognized by the antibody [2,53]. For instance, S-IgA directed against the O-antigen induces bacterial outer membrane distortion [36,53]. Second, the hydrophily of IgA Fc parts, and even more of the oligosaccharide side chains of SC, favours the entrapment of microorganisms in mucus [36]. Third, muco-ciliary clearance helps to evacuate bacterial clumps [54]. It must be emphasized that immune exclusion can also occur in the submucosa, where d-IgA forms immune complexes with subepithelial trespassing pathogens that are then transcytosed after binding to pIgR [55].

In addition to immune exclusion, S-IgA can directly block toxins and pathogens from adhering to the epithelium. For instance, reovirus type 1 (strain Lang) receptor-binding domain σ1 is recognized by IgA mAbs, preventing its binding to the epithelial cells [36,53]. Moreover, S-IgA may have a direct effect on bacterial virulence [53], while immune neutralization by IgA may also occur within the epithelial cell itself. According to in vitro studies with the measles virus [56], influenza virus [57], and human immunodeficiency virus [58], inactivation of the virus by specific IgA antibodies may even occur during the transcytosis.

In addition to neutralizing properties, IgA and S-IgA also regulate immune responses through the binding of specific Fc-receptors on myeloid leukocytes. FcαRI, also known as CD89, is indeed expressed by neutrophils, eosinophils, monocytes, macrophages, and dendritic cells [59]. In case of pathogen penetration through the mucosal barrier, cross-linking of IgA bound to pathogens or immune complexes to FcαRI can activate immune mechanisms such as phagocytosis of IgA-opsonized particles, release of cytokines and activated oxygen species, as well as antibody-dependent cell-mediated cytotoxicity [36,39,60]. IgA binding to FcαRI also activates human eosinophils [2], whereas S-IgA activates eosinophil degranulation even more efficiently, possibly by binding to a C-lectin-type SC receptor [2,61]. In addition, S-IgA is recognized by DC-SIGN, a lectin receptor mainly expressed by dendritic cells [36]. In mice, interaction between S-IgA and DC-SIGN homolog induced tolerogenic responses in dendritic cells, promoting the expansion of regulatory T cells (Treg), indicating that S-IgA may exert immunoregulatory properties [36,60]. In addition, in vitro studies have shown that polymeric IgA is able to activate the alternate complement system through the binding of mannose binding lectin [55,62]. In contrast, S-IgA is unable to bind and activate the classical complement pathway, which may highlight that S-IgA is primarily designed to prevent highly inflammatory reactions [63,64].

Finally, SC also has specific functions. First, it enhances S-IgA stability and resistance against proteolysis as compared with d-IgA [51]. Second, SC increases S-IgA hydrophilicity through N- and O-glycosylation, favouring its diffusion across the mucus layer [51]. At mucosal surfaces, both free and bound SC (in S-IgA) have been shown to alter *Streptococcus pneumoniae* [65] and *Escherichia coli* attachment to the epithelium [36,53,66], demonstrating SC-driven anti-microbial properties. Moreover, hydrophilic SC glycosylation contributes to SC protective abilities [67], while S-IgA interacts with various commensal bacteria through Fab- and Fc-independent mechanisms where SC plays a critical role, as SC deglycosylation also alters the interaction between S-IgA and Gram-positive bacteria, according to confocal microscopy [66]. Finally, SC is also thought to display anti-inflammatory properties, including neutralization of IL-8/CXCL-8 activity, thereby limiting the recruitment of neutrophils at mucosal surfaces [51].

### 2.3. Regulation of S-IgA Production

Although more extensively studied in the gut, the pIgR/IgA system regulation mechanisms in the airways have received increasing attention in the past decade, drawing a global picture where it relies on the complex interactions of immune, environmental, microbial, as well as hormonal factors.

First, the *PIGR* gene promoter displays binding sites for inflammation-related factors such as IFN regulatory factor 1 (IRF-1), STAT6 and Nuclear Factor (NF)-κB [52]. Therefore, host cytokines that activate pathways involving STAT, IRF or NF-κB, such as TNF-α, IFN-ɣ, IL-4 and IL-1, are able to upregulate pIgR expression and d-IgA transepithelial routing [51,52,68]. Depending on the studies, however, exposure to inflammatory stimuli provides divergent results. For instance, IL-4 may stimulate pIgR expression in Calu-3 cell line cultures [69], while it inhibits pIgR expression in primary airway epithelial cells [70], contributing to pIgR downregulation found in the airway epithelium of asthma patients. A similar dual effect in cell line versus primary cells was observed with TGF-β_1_, as exogenous exposure of Calu-3 cells to TGF-β_1_ increases SC production, whereas pIgR production is inhibited by TGF-β_1_ in primary human bronchial epithelial cells [68,71]. The molecular substratum for such discrepancies remains unknown. In addition, inflammatory cytokines contribute to pIgR downregulation both in asthma and COPD, while IL-17 conversely upregulates pIgR in *Pseudomonas aeruginosa* (*Pa*) infected CF cells [68,70,72,73].

Finally, environmental factors may also influence pIgR expression. For instance, pIgR mRNA levels are increased in ex-smokers’ lungs. This increase is however not observed at the protein level, or in situ nor in air/liquid interface cultures of primary human bronchial epithelial cells exposed to cigarette smoke (CS) or derived from smokers [68,74]. These data suggest that the CS exposure acts as a player in the regulation of pIgR gene expression in vivo, which is further submitted to post-transcriptional modifications that have not been much explored so far. In addition, the microbiota also regulates pIgR through the release of microbe-associated molecular patterns (MAMPs), which control *PIGR* gene transcription through the activation of Toll-like receptors (TLR) [51,52].

## 3. The Mucosal S-IgA System in Airway Disease

Figure 2 summarizes the mechanisms by which the IgA/pIgR system is altered in chronic respiratory diseases. Although resulting from complex physiopathological mechanisms in these diseases, respiratory tract colonization by pathogens and trespassing of the mucosal barrier have been shown to trigger these diseases, demonstrating their contribution to the development of such diseases [75,76].

### 3.1. COPD

The functionality of the IgA/pIgR system in respiratory diseases was first explored in COPD, where the abundant literature now clearly demonstrates its multifaceted alteration [77]. Chronic obstructive pulmonary disease (COPD), currently representing the third leading cause of death worldwide [3], is mainly due to CS with potential additional contributions of other toxics (biomass, occupational, air pollution) and genetic predisposition [78]. It is characterized by a progressive and mostly irreversible airway obstruction related to small airway pathology and destruction of the alveolar walls, referred to as emphysema [79]. Its clinical course can be dotted with acute worsening of respiratory symptoms, called exacerbations, a substantial proportion of which are driven by respiratory pathogens [5,8,80].

In 2001, our team first showed that pIgR/SC expression is decreased in the epithelium of large and small airways from COPD patients, as compared with both non-smokers and non-COPD smokers. In addition, SC expression inversely correlates with COPD severity and related functional parameters such as FEV1, FVC and MEF_50_ [81]. These observations were later corroborated in a larger cohort demonstrating that pIgR epithelial expression is mainly decreased in the airway epithelium from severe COPD patients, as compared with non-smoker controls, non-COPD smokers and less severe COPD patients, and associated with a persistence of the defect in primary cultures of bronchial epithelial cells from such patients where the mechanism could be shown as involving TGF-β signalling [68]. Recently, a genome-wide association study showed that *PIGR* gene expression was reduced in airway intermediate and ciliated cells from smokers without COPD [82]. Interestingly, pIgR downregulation in COPD is more obvious in zones of bronchial epithelial remodelling, such as goblet cell hyperplasia, squamous metaplasia, or incompletely differentiated areas. Accordingly, these specific zones also display S-IgA deficiency [83], contributing to localized reduced mucosal immunity, with those areas also exhibiting increased bacterial invasion, macrophage and neutrophil infiltration, as well as NF-κB activation [84].

Data regarding S-IgA concentrations in lung-lining fluids in COPD are more scattered. While a study reported no difference in S-IgA BALF concentrations in COPD patients versus controls [85], another showed decreased S-IgA levels in BALF from COPD patients [83], while a third found increased sputum SC concentrations in COPD [80]. Aside from showing COPD-related decreased S-IgA concentrations at the surface airway epithelium level, Du et al. showed that the S-IgA production was nevertheless quite preserved in COPD at the level of submucosal glands, although probably not compensating for surface epithelium S-IgA deficiency [86]. Interestingly, according to a very recent and as yet not fully published study, submucosal glands could represent a survival niche for IgA-secreting plasma cells [87]. In addition, increased levels of subepithelial IgA_1_ (but not IgA_2_) have been observed in COPD following upregulated IgA_1_ production by lung B cells, including in lymphoid follicles [88]. Globally, these studies depict a situation with increased local IgA production and reduced d-IgA transcytosis at the surface airway epithelium level, with secondary IgA_1_ subepithelial accumulation in the airways. In addition to impaired secretion, proteolytic degradation by IgA1 proteinases released by pathogens such as *Pa* that may infect COPD airways [89,90,91], particularly in patients with severe disease and/or bronchiectasis, may also contribute to the decrease in airway S-IgA in this disease.

Apart from pIgR downregulation in situ, in COPD airway tissues, we also showed that the airway epithelium redifferentiated in vitro in air/liquid-interface displays decreased pIgR expression and SC apical release, as well as impaired IgA transepithelial routing of d-IgA [68]. More recently, we demonstrated that the dysregulation of the pIgR/SC system in COPD is durably imprinted in the COPD airway epithelium, as long-term cultures (up to 10 weeks) of ALI-redifferentiated COPD airway epithelial cells exhibit persistent decreases in pIgR expression and SC release, along with impaired pIgR mRNA levels [74], corroborating previous findings obtained at 4 weeks of ALI culture [83]. Interestingly, increased IL-6 and BAFF/APRIL-TACI signalling have been observed in COPD at the epithelium level, driving class switch recombination towards IgA [88]; a feature that was also observed in smokers without COPD [92]. This B-cell immune response, also underlying IgA upregulation in peribronchiolar lymphoid follicles [93], probably relates to toxic and microbial exposures occurring in the COPD lung and could be considered as an attempt to compensate for local IgA deficiency, which is partly inhibited by CS [88].

To further explore the role of pIgR in COPD, pIgR^−/−^ mice, which display no S-IgA (or S-IgM) in mucosal secretions [94], have been used. Interestingly, pIgR^−/−^ mice exhibit increased susceptibility to mycobacterial infections [95] and spontaneously develop COPD-like small airway and parenchymal remodelling upon aging (86). When compared with wild-type mice, they show increased airway wall thickness and emphysema. They also display increased numbers of alveolar neutrophils and macrophages, increased concentrations of neutrophil elastase and matrix metalloproteinase-12, increased NF-κB p65, and increased microbial invasion. Moreover, their inflammatory response to non-typeable Haemophilus influenzae lysates is upregulated, and this is modulated by S-IgA. In addition, CS-exposed pIgR^−/−^ mice were more prone to developing emphysema and airway remodelling upon CS exposure than wild-type mice. Strikingly, the authors finally showed that knockout mice bred in germ-free conditions were protected from developing age-related COPD-like features, demonstrating the role of the lung microbiome in COPD pathogenesis [96].

To the best of our knowledge, no study has yet directly explored whether local bronchial IgA deficiency correlates to the exacerbation rate in COPD. Nevertheless, data obtained from the observational SPIROMICS cohort study showed that COPD patients with severely reduced serum IgA levels (≤70 mg/dL) experienced more frequent and more severe exacerbations [11].

### 3.2. Asthma

As with COPD, asthma is a broad clinical entity that manifests with symptoms of wheezing, shortness of breath, cough, and chest tightness. Asthma is characterized by reversible airflow obstruction, in contrast with COPD where the airflow obstruction is (mostly) irreversible [97]. Behind this clinical definition, asthma is a heterogeneous disease that relates to diverse pathophysiological mechanisms. Two main immune phenotypes are described, namely type 2 (T2-) high asthma, which includes several subsets according to the age of onset, allergic background, and comorbidities such as aspirin sensitivity and/or nasal polyposis; T2-low asthma, which includes obesity-related asthma [97,98,99,100]. Both early-onset allergic (or extrinsic) asthma and some forms of late-onset non-allergic asthma are characterized by a T2-mediated eosinophilic inflammation, whose underlying upstream mechanisms include epithelial activation by viruses, allergens and/or air pollution. The subepithelial penetration of these irritants is further enhanced by epithelial barrier dysfunction that is due to the decreased tightness of apical junctional complexes, as well as zones of epithelial shedding [77]. These airborne stimuli induce epithelial cells necrosis leading to the release of alarmins such as TSLP, IL-33, IL-25 and GM-CSF, the latter attracting and/or activating mast cells, basophils and eosinophils [51]. In T2-low asthma, macrophages recruitment is also driven by the epithelial release of CCL2 and CCL3 [101], while in obesity-related asthma, leptin activates airway epithelial cells and induces the production of cytokines such as IL-8, further contributing to neutrophil recruitment [101]. In turn, these inflammatory cells mediate airway remodelling, including goblet cells metaplasia, epithelial-to-mesenchymal transition and reticular basement membrane thickening, along with bronchial hyperresponsiveness and smooth muscle hypertrophy [51,98], as well as IgE class switch recombination in B cells [51]. In addition the increased levels of allergen specific IgE, higher IgA levels have been observed in the airway mucosa of allergic asthma patients [102,103,104]. In addition, class switch recombination of mucosal activated B cells towards IgA can be induced by different stimuli involved in an allergic response, such as double stranded RNA viruses in a TLR3-dependent pathway [105], or cytokines such as IL-4, which is increased in an asthmatic response [106] or by microbial stimuli through TGF-β production by dendritic cells [107].

Although the exact role(s) of S-IgA in the development of allergic disorders remains controversial, several lines of evidence suggest that the S-IgA system could play a protective role in asthma and allergic diseases [2]. Thus, higher levels of S-IgA in breast milk were associated to a lower risk of atopic dermatitis up to the age of two [108], and subjects with selective IgA deficiency display higher risks of developing allergic diseases [109]. In addition, populations with higher serum IgA levels display lower rates of house dust mite sensitization and severe airway hyperresponsiveness [110]. Conversely, a positive correlation exists between serum IgA specifically directed against cow’s milk β-lactoglobulin and wheat gliadin at the age of one, and IgE sensitization at the age of 6 [111].

Several mechanisms could explain these clinical observations. Firstly, S-IgA possibly mediates the immune exclusion of allergens, limiting the exposure of the immune system to these triggers [12,109]. It also interacts with several microbiota components, therefore modifying the composition of the microbiome, and influencing the maintenance of immune homeostasis in the gut and the lungs [112]. The alteration in the microbiota could increase the susceptibility to allergic disorders, as early-life antibiotic-treated mice displaying microbiota dysregulation showed a higher risk of developing allergic asthma [113,114]. Secondly, S-IgA seems able to drive tolerogenic immune mechanisms in dendritic cells through its binding to the DC-SIGN receptor [115]. Thus, the ligation of S-IgA to dendritic cells inhibits CD4^+^ T cell responses, in a DLL4-Notch pathway-dependent manner [116], while inducing a secondary expansion of Tregs [2]. This S-IgA-driven Treg population expansion could play a role in maintaining mucosal tolerance by regulating the excessive function of other cells and hampering aberrant T helper 2 responses observed in asthma [117]. However, this T helper 2-mediated inflammation is also enhanced by TSLP [118], which has also been described as able to inhibit both the generation of inducible Tregs [119,120] and the production of IgA [121].

In contrast, S-IgA can also trigger eosinophil degranulation [2,122] through its binding to FcαRI or to a putative C-lectin type SC receptor [123], leading to the release of IL-4 and IL-5, as well as eosinophil peroxidase and eosinophil cationic protein, thereby promoting Th2 inflammation and asthma pathogenesis and exacerbations [124]. Soluble S-IgA also enhances eosinophil survival in vitro [125]. FcαRI is also expressed by neutrophils, but conflicting data have been obtained regarding the pro- or anti-inflammatory effects of S-IgA binding to neutrophils. Indeed, myeloma-derived IgA alters neutrophil chemotaxis in vitro [126], while FcαRI binding has also been reported to increase the release of several cytokines by neutrophils [127].

Finally, previous findings described decreased S-IgA levels in BAL from symptomatic asthma patients. We also observed a dysregulation of the S-IgA/pIgR system in the airways in allergic asthma, that was reproduced in vitro by exposing the air-liquid interface cultures of human bronchial epithelial cells to exogenous IL-4 and IL-13, underlining the role of T2 inflammation in this impairment [70].

### 3.3. Cystic Fibrosis

In contrast with asthma and COPD [128], data concerning IgA in cystic fibrosis (CF) are scarcer. CF is a multisystemic disease mainly affecting the respiratory, digestive and reproductive tracts [129]. Chronic pulmonary infections, notably due to *Pa*, play a crucial role in the prognosis of the disease and constitute a major immune challenge for the airway epithelium [130]. Recently, we undertook one study assessing the pIgR/IgA system in a multimodal project, showing increased epithelial IgA^+^ B cells, IgA production and pIgR expression in the airway tissues, sputum, and serum from CF patients [72,73]. In addition, a mice model mimicking chronic lung infection by using *Pa*-coated microbeads instilled in the mice airways showed that infection could upregulate pIgR expression and IgA production in the lungs of F508del mice, partly in an IL-17-dependant manner [72]. This was observed in vivo although in vitro, CF-derived airway epithelial cells displayed reduced pIgR expression, that appeared to be related to the activation unfolded protein response. In parallel, pIgR transcytosis is enhanced in CF, as evidenced by increased SC concentration in CF sputum, although SC seems to be dysfunctional in CF, notably preventing its neutralization capacity of IL-8/CXCL-8 due to an altered glycosylation pattern [131]. Increased IgA concentration has been found in the serum from CF patients, as compared with controls [132], in particular in chronically *Pa* infected patients [133] or with high clinical severity according to Shwachman Kulczycki scores [134]. Similarly, increased IgA concentrations were found in BALF from CF patients as compared with controls [135]. In addition, increased specific anti-*Pa* IgA levels were observed in the serum of infected CF patients [136,137,138] and in patients with poor lung function, witnessed by low vital capacity and forced expiratory volume in one second [139], but also in sputum [140] and nasal lavage [141]. Finally, data regarding IgA secretion in the digestive and respiratory systems in CF diverge from results in the respiratory tract, as decreased IgA secretion in CF saliva [136] and gastric luminal perfusate have been reported [142], while an older study reports decreased free SC concentration in the sputum and saliva from CF patients [143]. It appears likely that in CF the IgA system is affected via multiple components and interplays between the host epithelium, immune cells, and microbiome that ultimately differently regulate IgA immunity at mucosal sites.

### 3.4. Bronchiolitis Obliterans Syndrome

Chronic lung allograft dysfunction (CLAD) represents a major hurdle in lung transplantation (LT), that burdens long-term survival [137]. Depending on pulmonary function tests and computed tomography, CLAD is currently separated into obstructive CLAD (first described as bronchiolitis obliterans syndrome (BOS), restrictive CLAD (also known as restrictive allograft syndrome (RAS), mixed CLAD (combining BOS and RAS features) and undefined CLAD [138]. BOS is the most frequent form of CLAD and is histologically characterized by epithelial injury, bronchocentric mononuclear inflammation, and fibrosis of small airways [144].

As repetitive and/or chronic infections have widely been associated with CLAD risk [145,146,147], it can be questioned whether impaired airway mucosal immunity (and thus reduced microbial eviction) could favor CLAD/BOS. However, the literature in the field is scarce, with no study so far assessing the airway expression of pIgR after LT. One previous study assessed BALF S-IgA levels in LT recipients, showing reduced S-IgA levels during so-called “rejection episodes” [148]. Of note is that, CLAD phenotypes had not been described at the time this study was published. More recently, Vandermeulen and colleagues showed increased total BALF IgA levels in LT recipients with RAS and BOS as compared with control LT recipients [149]. Finally, low serum IgA levels prior to LT represent a risk factor for developing post-LT infections [150,151], while lower post-LT IgA serum levels have been observed in patients with BOS as compared with control LT recipients [151].

## 4. Conclusions and Perspectives

Recent research demonstrates that the S-IgA system is altered during the course of chronic airway diseases, primarily in relation to a generic mechanism of the downregulation of pIgR expression that is observed upon airway epithelial dedifferentiation. Some mechanisms might prevail in certain disease phenotypes, such as the aberrant reactivation of developmental pathways (e.g., TGF-β and WNT/β-catenin, in COPD), immune activation through the IL-4R (e.g., in asthma) or unfolded protein response (e.g., in CF). In contrast, the production of IgA by B cells is upregulated, showing that the epithelial transport of IgA also represents the limiting factor for IgA immunity in the diseased lung. This epithelial pIgR defect may also lead to increased susceptibility to COPD development, through complex mechanisms involving aging, inflammation and the microbiome as shown in pIgR KO mice, as well as increased susceptibility to lung infections.

A better understanding of IgA mucosal immunity in the airways could deliver therapeutic perspectives. First, passive immunotherapy could be studied by using either specific IgA, as shown in a preclinical model of an airway allergy to ragweed pollen [152], or human polyclonal IgA as achieved with IgG for IgG deficiency. This later case is however not supported by a specific indication as it is known that selective IgA deficiency which represents the most frequent immunodeficiency, very rarely leads to opportunistic and/or frequent infections when present alone (as opposed to combined IgA and IgG deficiency) [153,154]. In contrast, supplementation with secretory IgA could be considered in diseases with evidence of the impaired transport of IgA into mucosa secretions such as asthma or COPD. Evidence of a benefit of such an approach is however lacking at both preclinical and clinical levels. For instance, in asthma, the putative benefit against infections might be at least theoretically counterbalanced by the detrimental activation of eosinophils. In this case of mucosal IgA deficiency, the approach could alternatively involve tackling the underlying mechanism within the epithelium, by interfering with the activation of IL-4/IL-13 (e.g., in asthma) or the TGF-β or WNT pathway (e.g., in COPD). Such an approach, however, implies considering targeted application (referring to the frequently opposed effects in the conducting epithelium versus alveolar epithelium) and careful evaluation of the side effects due to the pleiotropic nature of these pathways in lung homeostasis.

Future research should, thus, appreciate whether current therapies affect the S-IgA system and whether supplementing IgA could provide clinical benefits in patients with COPD or other chronic airway diseases.

## Figures and Tables

**Figure 1 cells-11-01324-f001:**
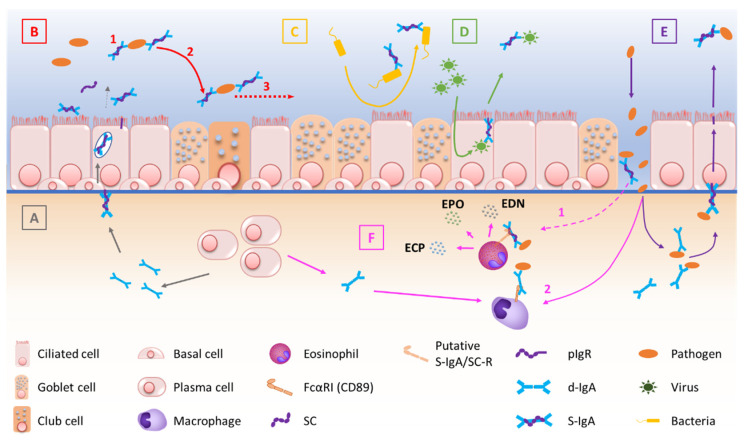
Schematic description of the multiple functions of IgA and the pIgR/S-IgA system at mucosal surfaces. (**A**) pIgR-mediated endosomal transcytosis of d-IgA, produced by submucosal B cells. (**B**) S-IgA-driven immune exclusion, including agglutination (1), entrapment of immune complexes (2) and clearance of trapped pathogens (3). (**C**) Inhibition of bacterial adherence to the mucosal epithelium. (**D**) Intraepithelial neutralization of penetrating viral antigens. (**E**) pIgR-mediated elimination of subepithelial immune complexes, after immune exclusion of pathogens by subepithelial d-IgA. (**F**) (1) S-IgA-induced degranulation of eosinophils resulting from binding of S-IgA to its eosinophil receptor (potentially FcαRI or another receptor such as C-type lectin), occurring possibly after epithelial barrier disruption and releasing eosinophil granule proteins. (2) IgA-induced engagement of phagocytes, enhancing clearance mechanisms.

**Figure 2 cells-11-01324-f002:**
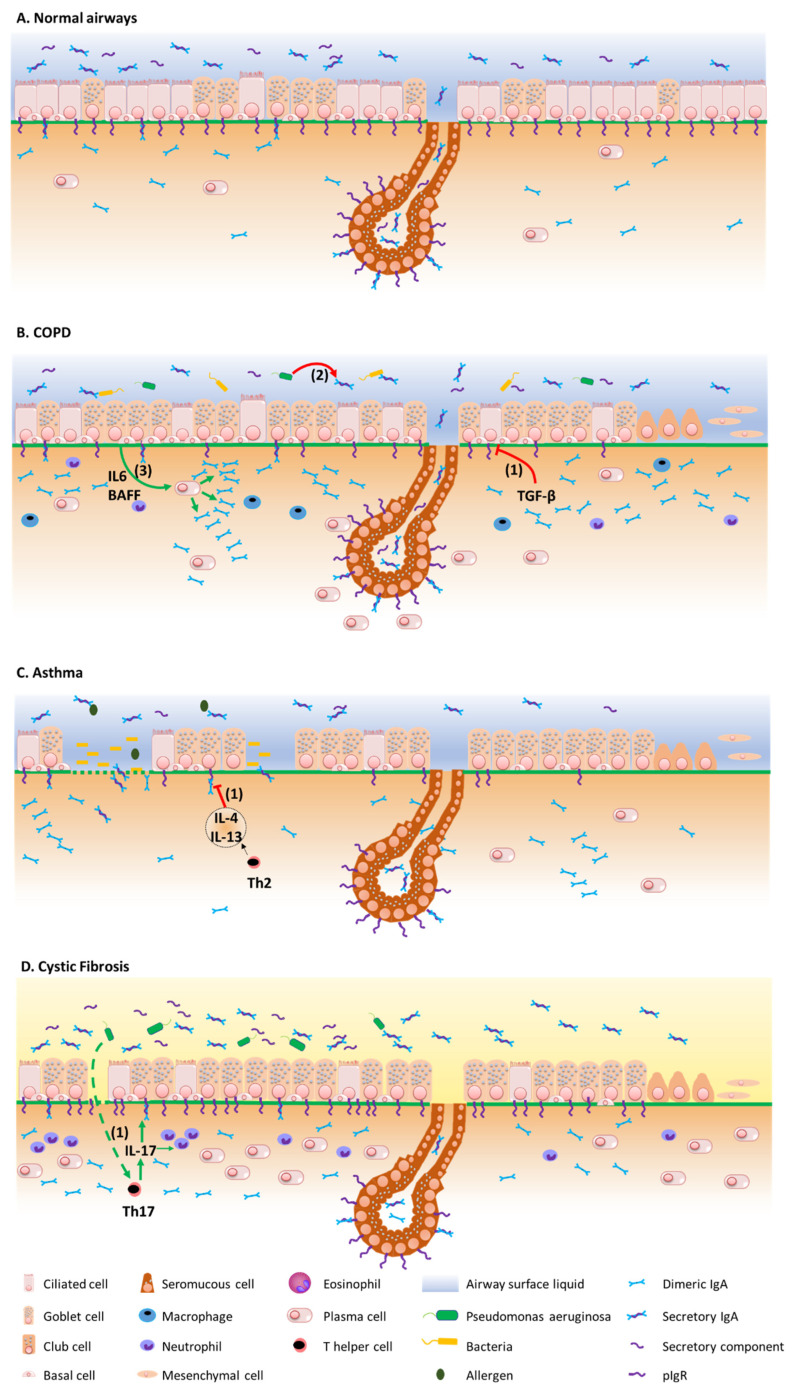
Overview of the pIgR/IgA system dysregulation mechanisms in chronic respiratory diseases. (**A**) pIgR/IgA system in airway homeostasis, at the epithelial surface and in submucosal glands. (**B**) In COPD, TGF-β induces pIgR downregulation at the epithelial surface (1), while pIgR expression is preserved in the submucosal glands. The S-IgA local deficiency relates to the subsequently decreased IgA transcytosis, as well as to S-IgA proteolysis by pathogen-derived proteinases (2), favouring bacterial invasion and innate immune cell infiltration. Subepithelial IgA may accumulate as a result of decreased transepithelial transport and IL-6- and BAFF-driven IgA synthesis (3). Enhanced survival of IgA+ plasma cells around the submucosal glands could contribute to a preserved S-IgA production at this level. (**C**) In asthma, IL-4/IL-13 may induce pIgR downregulation, also leading to luminal S-IgA deficiency (1). (**D**) In CF, pIgR is conversely upregulated, along with increased production of IgA and S-IgA in airway tissues and lumen, possibly through chronic infection by *Pseudomonas aeruginosa* that drives pIgR upregulation through IL-17 (1).

## Data Availability

Not applicable.

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
