# Peer review of "Secretory Immunoglobulin A Immunity in Chronic Obstructive Respiratory Diseases"

_cells, 2022, doi:10.3390/cells11081324_

Round 1

Reviewer 1 Report

The introduction section is very poorly written and also in a very biased way. No information about incidence or prevalence of chronic obstructive respiratory diseases in human populations, the age of affected people, etc. No information about the role of other important first-line mucosal protective factors (mucus, cilia, antibacterial enzymes, etc). No information about types of mucosa and differences among them.

It is very important to briefly summarize the actual situation in the human population, including the age of affected people, and to present all factors involved in these diseases, including the first line of mucosal protection. And then there is also innate and adaptive immunity, in addition to mucus and epithelial barrier.

Section 2: I had expected to get the necessary information about the S-IgA system in homeostasis in respiratory mucosa but instead the whole section on 6 pages is only summarizing the literature on S-IgA mostly in the digestive system (which is 2.1. Production and structure of S-IgA, 2.2. Functions of S-IgA, 2.3. Regulation of S-IgA production fro page 2 to page 7). In addition, the types, subtypes, and various forms of IgA as well as the production of IgA are already very difficult to understand. In a form written by the authors, it is very difficult to follow the subject, since the authors very quickly jump from one to another mucosa – mixing everything. It is very difficult to follow already for an expert from the field, but impossible for a general reader. I strongly recommend reforming this section to be more readable and user-friendly and perhaps adding the figures to improve readability.

Although SIgA is important in all mucosal surfaces, their production and role are specific for particular mucosal surfaces (respiratory, gastrointestinal, genitourogenital) thus, generalization is misleading and does not contribute to improved knowledge.

Why do the authors talk about the regulation of SIgR production in the gut if the review is about COPD and respiratory diseases? The mucosa in the respiratory tract differs from those in the gut, as well as the function, microorganisms, and diseases!

If the authors talk about the SIgA in general – do it in a separate section – but be aware that there are many very good and reader-friendly reviews on this subject. Therefore, I would not recommend it – it is better to refer the reader to the excellent review articles and include additional Figures about the molecular mechanisms (biology and production of SIgA). You can briefly explain the mechanism in the Figure text. It would be more helpful. In the manuscript, a section about what is actually known in respiratory mucosa, including nasal and oral mucosa is necessary (Figure is also recommended). The authors do not mention nasal or oral mucosa at all. Why? So, the manuscript from the start to page 8 is very confusing and misleading. In addition, I wonder why more results of the studies in humans are not included in this part.

Summarizing literature from all mucosas without a critical mind is counterproductive and useless for the reader. There is a question, what is the meaning of such a review? Section 3 is interesting and better written (although it is written in a biased way as well because it does not mention all necessary information about a particular disease). In general, it appears that sections 1 and 2  in the manuscript do not offer the necessary basic knowledge to follow the information in section 3. To understand section 3, Sections 1 and 2 should provide an introduction of the current knowledge about the respiratory mucosa (including nasal and oral), and the biology, function, and production of SIgA in the respiratory mucosa. Basic histological overview of the respiratory tract to explain the first line protective function in respiratory mucosa (i.e. the cells involved in the production of SIgA in the respiratory mucosa, the role of lamina epithelialis, lymph nodes, etc. ). Namely, this information is of crucial importance to understanding the role of SIgA in respiratory disease.

There are already very good review articles on the biology of SIgA in general. The manuscript in the current form does not reach the standards for publication. The review is not in accordance even with the title and the aims of the article.

Author Response

The introduction section is very poorly written and also in a very biased way. No information about incidence or prevalence of chronic obstructive respiratory diseases in human populations, the age of affected people, etc. No information about the role of other important first-line mucosal protective factors (mucus, cilia, antibacterial enzymes, etc). No information about types of mucosa and differences among them.

It is very important to briefly summarize the actual situation in the human population, including the age of affected people, and to present all factors involved in these diseases, including the first line of mucosal protection. And then there is also innate and adaptive immunity, in addition to mucus and epithelial barrier.

  1. General information regarding demographics of COPD, asthma, and CF has been added in the introduction. In addition, a brief summary of other mucosal protection mechanisms has been added. However, it must be kept in mind that the pIgR/IgA system is the main focus of the review, so we did not elaborate on these other mechanisms.

Section 2: I had expected to get the necessary information about the S-IgA system in homeostasis in respiratory mucosa but instead the whole section on 6 pages is only summarizing the literature on S-IgA mostly in the digestive system (which is 2.1. Production and structure of S-IgA, 2.2. Functions of S-IgA, 2.3. Regulation of S-IgA production fro page 2 to page 7). In addition, the types, subtypes, and various forms of IgA as well as the production of IgA are already very difficult to understand. In a form written by the authors, it is very difficult to follow the subject, since the authors very quickly jump from one to another mucosa – mixing everything. It is very difficult to follow already for an expert from the field, but impossible for a general reader. I strongly recommend reforming this section to be more readable and user-friendly and perhaps adding the figures to improve readability.

Although SIgA is important in all mucosal surfaces, their production and role are specific for particular mucosal surfaces (respiratory, gastrointestinal, genitourogenital) thus, generalization is misleading and does not contribute to improved knowledge.

Why do the authors talk about the regulation of SIgR pIgR production in the gut if the review is about COPD and respiratory diseases? The mucosa in the respiratory tract differs from those in the gut, as well as the function, microorganisms, and diseases!

  1. A brief comparison between nasal, bronchial and digestive mucosae is now provided. Also, we tried to regroup data from the digestive tract at the end of the paragraphs. Nevertheless, this section is dedicated to mechanisms that regulate pIgR expression and IgA production, that have been widely studied in the gut, with relevance to the respiratory tract and chronic respiratory diseases even if both tracts differ in many ways.

If the authors talk about the SIgA in general – do it in a separate section – but be aware that there are many very good and reader-friendly reviews on this subject. Therefore, I would not recommend it – it is better to refer the reader to the excellent review articles and include additional Figures about the molecular mechanisms (biology and production of SIgA). You can briefly explain the mechanism in the Figure text. It would be more helpful. In the manuscript, a section about what is actually known in respiratory mucosa, including nasal and oral mucosa is necessary (Figure is also recommended). The authors do not mention nasal or oral mucosa at all. Why? So, the manuscript from the start to page 8 is very confusing and misleading. In addition, I wonder why more results of the studies in humans are not included in this part.

  1. Since this review focuses on respiratory diseases, we did not add a full section on pIgR/IgA immunity in oral and nasal mucosa, that has moreover been summarized in recent reviews. It would induce confusion and unnecessarily elongate the manuscript. Occasional data from the gut mucosa are merely highlighted to complete respiratory data.

Summarizing literature from all mucosas without a critical mind is counterproductive and useless for the reader. There is a question, what is the meaning of such a review? Section 3 is interesting and better written (although it is written in a biased way as well because it does not mention all necessary information about a particular disease). In general, it appears that sections 1 and 2  in the manuscript do not offer the necessary basic knowledge to follow the information in section 3.

  1. These points are already addressed in response 1.

To understand section 3, Sections 1 and 2 should provide an introduction of the current knowledge about the respiratory mucosa (including nasal and oral), and the biology, function, and production of SIgA in the respiratory mucosa. Basic histological overview of the respiratory tract to explain the first line protective function in respiratory mucosa (i.e. the cells involved in the production of SIgA in the respiratory mucosa, the role of lamina epithelialis, lymph nodes, etc. ). Namely, this information is of crucial importance to understanding the role of SIgA in respiratory disease.

There are already very good review articles on the biology of SIgA in general. The manuscript in the current form does not reach the standards for publication. The review is not in accordance even with the title and the aims of the article.

  1. As previously mentioned, we globally tried to improve sections 1 and 2 to allow the reader to understand better the 3rd section.

Reviewer 2 Report

With interest, I read the manuscript cells-1611509.

It is a comprehensive review written by experiences researchers, who are well-recognized in the field.

I have minor and/or facultative comments only:

  1. Please, be careful with the details. E.g. in line 89, some words are underlined.
  2. All abbreviations need to be explained upon their first appearance, which is not always the case, e.g. TGF-β.
  3. Figures 1 and 2 are scientifically comprehensive and graphically nice. However, I am very much afraid that some details (e.g. ECP, EPO, IDN in Figure 1 or Th2 or Th17 in Figure 2) may not be visible without a substantial magnification. Could those be made larger?
  4. All abbreviations used in the figures need to be explained in their legends.
  5. 2. Asthma. Asthma pheno-/endotypes. Also obesity-associated asthma should be mentioned in one sentence (PMID: 32973742).
  6. 2. Asthma. Asthma pheno-/endotypes and their mechanisms. The contribution of airway epithelium should be expanded to 2-3 sentences (PMID: 31904412).

Author Response

With interest, I read the manuscript cells-1611509.

It is a comprehensive review written by experiences researchers, who are well-recognized in the field.

We thank the reviewer for their interest in our manuscript and for their comments.

I have minor and/or facultative comments only:

  1. Please, be careful with the details. E.g. in line 89, some words are underlined.

The manuscript was carefully reviewed to correct these details.

  1. All abbreviations need to be explained upon their first appearance, which is not always the case, e.g. TGF-β.

A table of abbreviations has been added at the end of the manuscript.

  1. Figures 1 and 2 are scientifically comprehensive and graphically nice. However, I am very much afraid that some details (e.g. ECP, EPO, IDN in Figure 1 or Th2 or Th17 in Figure 2) may not be visible without a substantial magnification. Could those be made larger?

The figures were modified accordingly.

  1. All abbreviations used in the figures need to be explained in their legends.

A table of abbreviations has been added at the end of the manuscript.

  1. 2. Asthma. Asthma pheno-/endotypes. Also obesity-associated asthma should be mentioned in one sentence (PMID: 32973742).

Obesity-associated asthma is not clearly mentioned, as suggested.

  1. 2. Asthma. Asthma pheno-/endotypes and their mechanisms. The contribution of airway epithelium should be expanded to 2-3 sentences (PMID: 31904412).

A few lines summarizing the role of airway epithelial activation in the development of asthma pheno-/endotypes have been added.

Reviewer 3 Report

COPD, asthma and cystic fibrosis are major respiratory diseases in the United States and mucosal immunity plays an important role in disease pathogenesis. The review article submitted by Charlotte de Fays provides a comprehensive, up-to-date description on secretory immunoglobulin A immunity and its involvement in respiratory disease development. The review is well written and reflects the newest knowledge on pIgR/IgA mucosal immunity. However, it would be great for authors to further comment on how the improved understanding of pIgR/IgA mucosal immunity contributes to the therapeutic development or respiratory disease treatment in the Conclusion and Perspective section.  

Author Response

COPD, asthma and cystic fibrosis are major respiratory diseases in the United States and mucosal immunity plays an important role in disease pathogenesis. The review article submitted by Charlotte de Fays provides a comprehensive, up-to-date description on secretory immunoglobulin A immunity and its involvement in respiratory disease development. The review is well written and reflects the newest knowledge on pIgR/IgA mucosal immunity. However, it would be great for authors to further comment on how the improved understanding of pIgR/IgA mucosal immunity contributes to the therapeutic development or respiratory disease treatment in the Conclusion and Perspective section.

We thank the reviewer for their interest in our manuscript and for their comments.

As requested, a few lines were added regarding the therapeutic alternatives that could be implemented in chronic respiratory diseases, in link with the pIgR/IgA system.

Round 2

Reviewer 1 Report

Dear authors, 

Although it seems that the authors made a huge reformation of their manuscript (red text), in reality, their corrections were only minor. The authors rejected presenting the information needed for the understanding of the subject. Therefore, I am struggling with the manuscript and I wonder what is the message of the manuscript? The title of the manuscript is »Secretory immunoglobulin A immunity in chronic obstructive respiratory diseases«. But in sections 1 and 2 I do not get the basic information needed for understanding section 3. Instead, in section 2 there are so many unnecessary details from all mucosas that I do not know what to do with them and I can not find the link between section 2 and section 3.

  • Section 2 summarizes numerous pieces of information on the mucosal S-IgA system in homeostasis in general. To be honest, the information described in section 2 is nothing new and represents more than half of the article. In fact, this information is described in numerous articles in a much better way. So, please reduce this section by at least half. I strongly suggest and insist on shortening this section and presenting it in a form of a picture, legend description, and table. A picture with the description in the legend would be more reader-friendly and contribute to a better understanding of the subject.
  • Instead of improving section 2 (reduction with schematic representation) and including an explanation about the production of IgA (not only S-IgA) in the respiratory tract (including nose and oral mucosa and lymph nodes) as recommended by the reviewer, in the revised version the authors rather included an oversimplified explanation of differences in the gut mucosa, which makes more harm than benefit and needs to be deleted. Please be aware that the manuscript is about respiratory tract diseases.
  • It is very important that the authors include the mucosa and immune system that is implicated in the generation of the IgA (not S-IgA) in the respiratory tract. The authors briefly mentioned the role of other important first-line mucosal protective factors (mucus, cilia, antibacterial enzymes, etc). However, the authors rejected including innate and adaptive immunity. Please be aware that IgA is a part of the immune system and many immune cells are involved in the production and function of IgA! The respiratory tract has its own protective mechanisms (starting in the nasal or oral cavity), which need to be sufficiently explained in the manuscript. The authors need to include an additional section to describe the implication of the immune system in the production of IgA (not only S-IgA) in the respiratory tract. From section 3 it is obvious that the production of IgA includes also the immune system. This information is of crucial importance to understanding the role of IgA and SIgA in respiratory disease and need to be sufficiently explained in the first part of the manuscript.
  • I strongly suggest the authors include all necessary information about the production of IgA, which is a prerequisite for the generation of S-IgA! Basic histological overview of the respiratory tract to explain the first line protective function in respiratory mucosa (i.e. the cells involved in the production of IgA and pIgR in the respiratory mucosa, the role of lamina epithelialis, lymph nodes, etc). For instance, in section 3.1 the authors explain that in patients with severe COPD »pIgR epithelial expression is mainly decreased in the airway epithelium« and is » more obvious in zones of bronchial epithelial remodeling, such as goblet cell hyperplasia, squamous metaplasia, or incompletely differentiated areas.« In the first part of the manuscript, the authors need to explain which cells are responsible for pIgR expression in the respiratory tract in normal conditions.

Author Response

We thank the reviewer for her/his comments, which are extensively addressed hereunder.

Comment 1. Although it seems that the authors made a huge reformation of their manuscript (red text), in reality, their corrections were only minor.

Response 1. As the manuscript was initially returned for minor revision, only minor revisions were indeed provided. Following major revisions suggested here, more extensive changes have been made in this second revision.

C2. The authors rejected presenting the information needed for the understanding of the subject. Therefore, I am struggling with the manuscript and I wonder what is the message of the manuscript? The title of the manuscript is “Secretory immunoglobulin A immunity in chronic obstructive respiratory diseases”. But in sections 1 and 2 I do not get the basic information needed for understanding section 3.

R2. We have accordingly adapted the manuscript in order to better introduce the topic of changes in IgA mucosal immunity during respiratory diseases. Section 1 constitutes a short introduction to the diseases that are covered in the review and provides general information on why the study of the pIgR/S-IgA system is relevant to those diseases. Section 2 summarizes current knowledge on what the pIgR/S-IgA system is and how it functions (including S-IgA functions and regulation of its production); this section 2 has been profoundly modified according to the following comments.

C3. Instead, in section 2 there are so many unnecessary details from all mucosas that I do not know what to do with them and I can not find the link between section 2 and section 3.

R3. Based on this comment, Section 2 has been substantially shortened in the revised manuscript (about 2200 words instead of 3000) by focusing on the information related to the airways and providing only the information that is needed to understand Section 3.

  • Section 2 summarizes numerous pieces of information on the mucosal S-IgA system in homeostasis in general. To be honest, the information described in section 2 is nothing new and represents more than half of the article. In fact, this information is described in numerous articles in a much better way.  So, please reduce this section by at least half. I strongly suggest and insist on shortening this section and presenting it in a form of a picture, legend description, and table. A picture with the description in the legend would be more reader-friendly and contribute to a better understanding of the subject.

The general description of the pIgR/S-IgA system is provided to give the reader the keys to understand Section 3, as required above. However, it has been substantially reduced in the revised version, notably by removing information on serum IgA.

Another figure has not been added, as it seems very optional to understand the topic and as figures on pIgR/S-IgA functions are already available (e.g., Carlier et al, Clin Exp Allergy 2016)

  • Instead of improving section 2 (reduction with schematic representation) and including an explanation about the production of IgA (not only S-IgA) in the respiratory tract (including nose and oral mucosa and lymph nodes) as recommended by the reviewer, in the revised version the authors rather included an oversimplified explanation of differences in the gut mucosa, which makes more harm than benefit and needs to be deleted.  Please be aware that the manuscript is about respiratory tract diseases.

We agree with the Reviewer that this part, initially added to address the first Reviewer’s comments, was somewhat out-of-the-scope and too summarized. This information related to the gut mucosa has thus been removed.

In addition, this review aims at recapitulating alterations of the pIgR/S-IgA system in chronic obstructive respiratory disease (i.e. COPD, asthma, CF and BOS) and therefore, we did not add paragraphs on nasal and oral mucosa, which would also be confusing to the reader.

  • C4. It is very important that the authors include the mucosa and immune system that is implicated in the generation of the IgA (not S-IgA) in the respiratory tract. The authors briefly mentioned the role of other important first-line mucosal protective factors (mucus, cilia, antibacterial enzymes, etc). However, the authors rejected including innate and adaptive immunity. Please be aware that IgA is a part of the immune system and many immune cells are involved in the production and function of IgA! The respiratory tract has its own protective mechanisms (starting in the nasal or oral cavity), which need to be sufficiently explained in the manuscript. The authors need to include an additional section to describe the implication of the immune system in the production of IgA (not only S-IgA) in the respiratory tract. From section 3 it is obvious that the production of IgA includes also the immune system. This information is of crucial importance to understanding the role of IgA and SIgA in respiratory disease and need to be sufficiently explained in the first part of the manuscript.

R4. We thank the Reviewer for this suggestion. A paragraph on B cells and how they produce IgA has been added in the revised manuscript (Section 2.1, lines 119-134).

  • C5. I strongly suggest the authors include all necessary information about the production of IgA, which is a prerequisite for the generation of S-IgA! Basic histological overview of the respiratory tract to explain the first line protective function in respiratory mucosa (i.e. the cells involved in the production of IgA and pIgR in the respiratory mucosa, the role of lamina epithelialis, lymph nodes, etc). For instance, in section 3.1 the authors explain that in patients with severe COPD »pIgR epithelial expression is mainly decreased in the airway epithelium« and is » more obvious in zones of bronchial epithelial remodeling, such as goblet cell hyperplasia, squamous metaplasia, or incompletely differentiated areas.« In the first part of the manuscript, the authors need to explain which cells are responsible for pIgR expression in the respiratory tract in normal conditions.

R5. Accordingly, the cell composition of the bronchial epithelium and the role of each cell type are now provided at the beginning of Section 2 (lines 66-87). Also, cell types that have been shown to express pIgR are now stated in this section of the revised manuscript.

Round 3

Reviewer 1 Report

The authors followed most of the reviewer's suggestions and improved their manuscript.